# Vulnerability of β-Thalassemia Heterozygotes to COVID-19: Results from a Cohort Study

**DOI:** 10.3390/jpm12030352

**Published:** 2022-02-25

**Authors:** Sotirios Sotiriou, Athina A. Samara, Konstantinos E. Lachanas, Dimitra Vamvakopoulou, Konstantinos-Odysseas Vamvakopoulos, Nikolaos Vamvakopoulos, Michel B. Janho, Konstantinos Perivoliotis, Christos Donoudis, Alexandros Daponte, Konstantinos I. Gourgoulianis, Stylianos Boutlas

**Affiliations:** 1Department of Embryology, Faculty of Medicine, School of Health Sciences, University of Thessaly, 41110 Larissa, Greece; sotiriousoti@yahoo.gr (S.S.); kostantinos753@gmail.com (K.-O.V.); micheljanho@live.co.uk (M.B.J.); kperi19@gmail.com (K.P.); 2Department of Public Health and Social Medicine, Koutlimpanio General Hospital, 41221 Larissa, Greece; klachanas@hotmail.com; 31st Neonatal Intensive Care Unit, “Agia Sophia” Children’s Hospital, 11527 Athens, Greece; gina_dimitra@hotmail.com; 4Department of Biology, Faculty of Medicine, University of Thessaly, 41110 Larissa, Greece; nvamvak@yahoo.com; 5Department of Obstetrics and Gynaecology, Faculty of Medicine, University of Thessaly, 41110 Larissa, Greece; xdonoudis@gmail.com (C.D.); daponte@uth.gr (A.D.); 6Department of Respiratory Medicine, Faculty of Medicine, University of Thessaly, 41110 Larissa, Greece; kgourg@med.uth.gr (K.I.G.); sboutlas@gmail.com (S.B.)

**Keywords:** COVID-19, β-Thalassemia, mortality, critical care, pandemic

## Abstract

**Background:** The assignment of mortality risk from SARS-CoV-2 virus (COVID-19) to vulnerable patient groups is an important step toward containment of the pandemic. **Methods:** A total of 760 patients with a positive molecular test for SARS-CoV-2 who were unvaccinated against COVID-19 were recruited between 1 January and 30 June 2021. Patients were grouped by age; sex; and common morbidities, such as atrial fibrillation, chronic respiratory disease, coronary disease, diabetes type II, neoplasia, hypertension and β-Thalassemia heterozygosity. As a primary endpoint, we assessed mortality risk from COVID-19, and as secondary endpoints, we considered clinical severity and need for Intense Care Unit (ICU) admission. **Results:** In multivariate analysis, male sex (*p* < 0.001, OR = 2.59), increasing age (*p* < 0.001, OR = 1.049), β-Thalassemia heterozygosity (*p* = 0.001, OR = 2.41) and chronic respiratory disease (*p* = 0.018, OR = 1.84) were identified as risk factors associated with mortality due to COVID-19. Moreover, male sex (*p* < 0.001, OR = 1.98), increasing age (*p* < 0.001, OR = 1.052) and β-Thalassemia heterozygosity (*p* = 0.001, OR = 2.59) were associated with clinical severity in logistic regression. Regarding ICU admission, the risk factors were identified as male sex (*p* = 0.002, OR = 1.99), chronic respiratory disease (*p* = 0.007, OR = 2.06) and hypertension (*p* < 0.001, OR = 5.81). **Conclusions:** An increased mortality risk from COVID-19 was observed for older age, male sex, β-Thalassemia heterozygosity and respiratory disease. Carriers of β-Thalassemia were identified as more vulnerable for severe clinical symptomatology, but there was no increased possibility for ICU admission. Readjustment of these findings to consider impacts of variant strains prevailing during the latest viral outbreak among vulnerable patient groups may offer timely relief from the pandemic.

## 1. Introduction

Over the last two years, the SARS-CoV-2 virus (COVID-19) pandemic has spread globally, affecting every country worldwide [1]. The identification of common comorbidities that increase the mortality risk by severe acute respiratory syndrome–coronavirus 2 (COVID-19) is an important first step toward morbidity and mortality risk containment from the pandemic. Early identification, consultation and intervention practices among vulnerable groups could reduce the morbidity rates and mortality risk from COVID-19.

Several risk factors attributing to mortality due to COVID-19, including patients’ demographic characteristics and common comorbidities, have been identified since the beginning of the pandemic. More specifically, increasing age and male sex increase the severity and mortality risk [2,3]. Furthermore, common comorbidities, including hypertension, diabetes mellitus and chronic obstructive lung disease, have also been reported as factors associated with an increased mortality in infected patients [4,5].

β-Thalassemia is the most common inherited single gene disorder in the world, with approximately 1.5% of the global population being heterozygous for β-Thalassemia [6]. In a recently published pilot study during the second wave of the pandemic, β-Thalassemia heterozygotes were identified as a group of patients vulnerable to COVID-19, with an increased mortality risk [7]. These findings have to be updated during the next pandemic waves with novel variants of the virus being dominant.

Herein, to consolidate our pilot observations [7], we analyzed considerably more COVID-19-positive and unvaccinated cases among these vulnerable groups, spanning a longer time period. Mortality risk, clinical severity and need for Intense Care Unit (ICU) admission were assessed as endpoints.

## 2. Methods

### 2.1. Settings

The present retrospective cohort study includes 760 consecutive patients unvaccinated against COVID-19 with a positive SARS-CoV-2 Real-Time Polymerase Chain Reaction (RT-PCR) molecular test. All participants were registered in the emergency department (ER) of the tertiary referral center in central Greece (University Hospital of Larisa), between 1 January and 30 June 2021. 

### 2.2. Participants and Study Design

A retrospective analysis was conducted of medical and laboratory records from consecutive patients registered in the ER of a tertiary referral hospital. A database was created based on medical history and laboratory tests of confirmed COVID-19-positive subjects. We examined the mortality risk of vulnerable patient groups with common morbidity symptoms, including β-Thalassemia heterozygotes. The course of non-hospitalized participants was followed by telephone interviews. 

The primary outcome of this study was the association of clinical and demographic variants with mortality due to COVID-19 in infected individuals. COVID-19 infection was the single common official cause of death for our study participants, as registered in hospital archives. Furthermore, clinical severity of symptomatology and need for Intense Care Unit (ICU) admission were considered as secondary endpoints.

Patient demographic characteristics of age; sex; and common morbidities, including atrial fibrillation, chronic respiratory disease, coronary disease, diabetes, neoplasia and hypertension were recorded. In addition, β-Thalassemia heterozygosity was assessed through laboratory tests and known medical history. We excluded the current smoking-status parameter from patient characteristics being studied, based on preliminary indications of notable absence of statistically significant correlations between smoking and mortality from COVID-19. 

### 2.3. Ethical Considerations

Experimental therapeutic protocols were not applicable in this study. All data were analyzed anonymously, using code numbers with respect to the patient’s privacy, and collected in the context of routine diagnostic and therapeutic procedures. Nevertheless, the study conformed to the Research and Ethical Committee guidelines of the University Hospital of Larisa.

### 2.4. Sample Estimation

Considering an estimated prevalence of 8% in our entire study population, a precision of ±3.5% and 95% confidence interval (CI), the minimum sample size required was calculated by a precision analysis, using Epi Info 7 [8]. A minimum study sample was set at 231 patients. 

### 2.5. Statistical Analysis

Analysis was carried out by using SPSS version 26.0 (IBM, Chicago, IL, USA). Categorical variables are described by using frequency and relative frequency. Continuous variables are described with means and standard deviation. Analysis of continuous variables was conducted by using the Mann–Whitney U test and Kruskal–Wallis test, since the assumption of normal distribution was violated. Data were checked for deviation from normal distribution, using the Shapiro–Wilk normality test. Categorical data were analyzed with the use of Chi-square test or Fisher’s exact test. Multivariate analysis was performed in the form of binary logistic regression. For all the analyses, a 5% significance level was set. 

## 3. Results 

A total of 760 patients were included in the study, of which 448 (58.9%) were male and 312 (41.1%) female, with a mean age of 62.21 (±16.42) years, ranging from 20 to 93. A total of 448 (58,9%) patients were male and 312 (41.1%) female. Overall, 189 study participants died, resulting in a mortality rate of 24.86%. 

Regarding mortality, in univariate analysis, male sex (*p* < 0.001, OR = 2.44), increased age (*p* < 0.001), atrial fibrillation (*p* < 0.001, OR = 2.44), chronic respiratory disease (*p* < 0.001, OR = 2.71), coronary disease (*p* < 0.001, OR = 2.11), hypertension (*p* < 0.001, OR = 2.77) and β-Thalassemia heterozygosity (*p* < 0.001, OR = 2.26) were associated with increased mortality due to COVID-19 (Table 1).

In logistic regression analysis, male patients were 2.6 times more likely to die than female patients (*p* < 0.001, OR = 2.59). Furthermore, older participants were more likely to die from COVID-19; moreover, every year of age increased the possibility to die by 4.9% (*p* < 0.001, OR = 1.049). Patients with underlying chronic respiratory disease had a 1.8-times increased mortality possibility than patients without respiratory disease history (*p* = 0.018, OR = 1.84). Interestingly, β-Thalassemia heterozygotes had a 2.4-times increased possibility of mortality compared to patients without the trait (*p* < 0.001, OR = 2.41) (Figure 1). There was no statistically significant association between COVID-19 attributed mortality and other comorbidities, such as atrial fibrillation (*p* = 0.058), hypertension (*p* = 0.243), coronary disease (*p* = 0.617), diabetes type II (*p* = 0.439) and neoplasia (*p* = 0.653) (Table 1).

Regarding severity of clinical symptoms, in univariate analysis, male sex (*p* < 0.001), increased age (*p* < 0.001), atrial fibrillation (*p* < 0.001), chronic respiratory disease (*p* < 0.001), coronary disease (*p* < 0.001), hypertension (*p* < 0.001) and β-Thalassemia heterozygosity (*p* < 0.001) were associated with increased severity of clinical symptoms attributed to COVID-19 (Table 2). Moreover, in logistic regression, there was almost double the possibility of severe clinical disease for male patients than female ones (*p* < 0.001, OR = 1.98). Older participants were more likely to have symptoms with increased severity (*p* < 0.001, OR = 1.052), with every year of age increasing the possibility of severe disease by 5.2%. β-Thalassemia heterozygosity was also identified as an independent risk factor for severe clinical symptoms of COVID-19 (*p* < 0.001, OR = 2.59) (Figure 2). 

When assessing ICU admission, in univariate analysis, male sex (*p* = 0.001), increased age (*p* = 0.005), chronic respiratory disease (*p* < 0.001), coronary disease (*p* < 0.001) and hypertension (*p* < 0.001) were associated with the need for ICU admission. In logistic regression, male patients had double the possibility for ICU stay than female ones (*p* = 0.002, OR = 1.99). Furthermore, chronic respiratory disease and hypertension were identified as independent risk factors for ICU admission due to COVID-19 (*p* = 0.007, OR = 2.06 and *p* < 0.001, OR = 5.81 respectively), with patients with hypertension being 5.8 times more possible to need ICU. 

## 4. Discussion

In the present study, we assessed the role of common comorbidities as independent risk factors for COVID-19 attributed mortality among 760 unvaccinated patients against SARS-CoV-2 during the first half of 2021. The current findings support earlier observations of strong statistical association between mortality due to COVID-19 and male sex, increased age, chronic respiratory disease and β-Thalassemia heterozygosity [3,5,6,7,8]. Patients with underlying chronic respiratory disease were included in the high-mortality-risk group of patients from the beginning of the pandemic [9].

β-Thalassemias are a group of hereditary autosomal recessive anemias caused by either reduced or complete absence of production of β-globin chains of the hemoglobin tetramer [10]. Data regarding potential effects of associated comorbidities in thalassemic patients with COVID-19 are limited. Recent studies reported that patients with β-Thalassemia have a chronic condition which may contribute to an increase in susceptibility to SARS-CoV-2 infection [11,12]. The underlying disease is associated with several comorbidities and complications of chronic transfusions, including heart failure, pulmonary hypertension, hypogonadism and diabetes, that may attribute to the susceptibility of this group of patients to COVID-19 infection [11]. 

In this context, it is necessary to investigate the potential role of the β-Thalassemia trait as a risk factor for morbidity and mortality by COVID-19. The compromised nature of response to stress that is inherent to asymptomatic or mildly anemic β-Thalassemia heterozygotes facilitates collective induction of innate immune receptor CD45, Toll-like receptor 4 and CD32 expression; reduces ability to produce oxidative bursts; and elevates membrane lipid peroxidation [13]. Borderline resistance of β-Thalassemia trait carriers to stress may explain the low threshold of COVID-19 symptoms required to begin treatment that appear with considerable time lag, require longer periods of hospitalization and ICU care and result in over twice the possibility of mortality due to COVID-19.

The vascular nature of COVID-19 symptoms leads to mortality from cardiovascular (CVS) failure [14], rendering CVS control a primary target during the pandemic. CVS control is exerted through classical renin-angiotensin system (RAS)-inducing vasoconstriction via renin processed angiotensin II (ATII) vasoconstrictors, and counter-balancing non-classical RAS-inducing vasodilation via ACE2 conversion of ATII or processing of ATI to AT 1–7 vasodilators [15,16]. SARS-CoV-2 inhibits ACE2 expression and deranges CVS homeostasis [16]. Current COVID-19 treatment strategies aim to suppress SARS-CoV-2 main protease activity, required to release active viral protein products [17] and induce ACE2 expression [16]. Thus, CVS control of COVID-19 positive vulnerable patient groups by statins, merits thorough consideration as a potential first-line treatment option.

Updated data of patients with adaptive evolutionary viral alterations may improve the representability of data collection and enhance the reliability of associated clinical findings. Future studies on COVID-19 mortality risk should address pregnancy and high-risk pregnancy, particularly for women with genotypic variations associated with early onset preeclampsia, such as variant TLR-4 alleles [18], and mutant angiotensin type I and type II receptor combination genotypes [19].

This new addition may have not only statistical explanations (more participants analyzed), but mainly biological explanations (longer time period of data collection). A possible biological explanation for the observed mortality boost of this new subgroup may rely on the enriched blend of viral variants that prevailed during the first half of 2021, rather than during its narrower last quarter of 2020 previously examined [3]. According to this explanation, the reliability of clinical observations related to mortality impact from COVID-19 depends on timing of participant data collection during the course of the pandemic.

The present study identified independent risk factors associated with mortality due to COVID-19. The most noteworthy association is confirmation of the susceptibility of individuals heterozygous for β-Thalassemia to COVID-19. However, prior to the appraisal of these results, several limitations should be considered. Data were collected retrospectively, depending on the availability and accuracy of data records, and information bias may have occurred. Furthermore, data were collected from a single tertiary hospital, and the generalization of our findings is limited.

Analogous studies on the distribution of mortality risk from COVID-19 among heterozygotes of less frequent abnormal hemoglobins [20] may help elucidate key mortality factors of this vulnerable asymptomatic group and are worth pursuing on both clinical and academic grounds.

## 5. Conclusions

We conclude that COVID-19 mortality is affected primarily by male sex, aging, β-Thalassemia trait and chronic respiratory disease/asthma, followed by atrial fibrillation, hypertension, coronary disease, diabetes and neoplasia. Timely accounts of these observations with vaccination against SARS-CoV-2 will assess the effect of vaccination status on mortality risk and facilitate early identification, consultation and treatment of COVID-19-susceptible cases for optimal pandemic control.

## Figures and Tables

**Figure 1 jpm-12-00352-f001:**
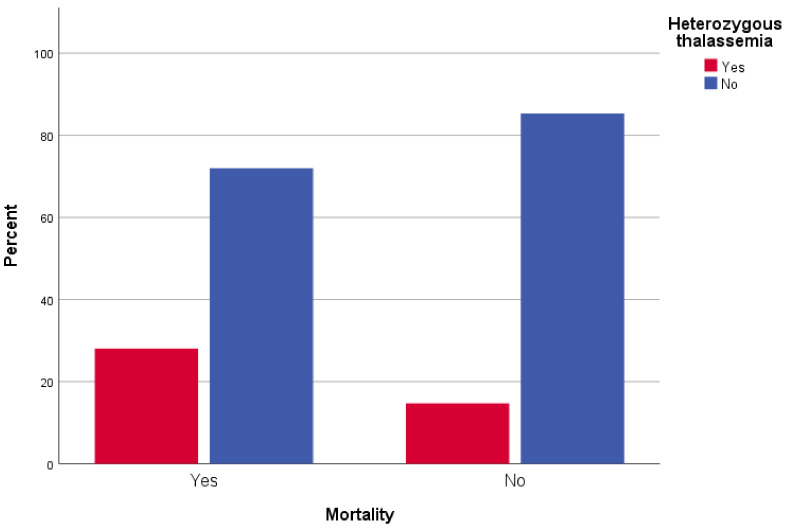
Mortality of β-Thalassemia heterozygotes from COVID-19: distribution of SARS-CoV-2 infected β-Thalassemia trait carriers (red box) among study participants who died (Yes), or survived (No) from COVID-19, relative to non-carriers (blue box).

**Figure 2 jpm-12-00352-f002:**
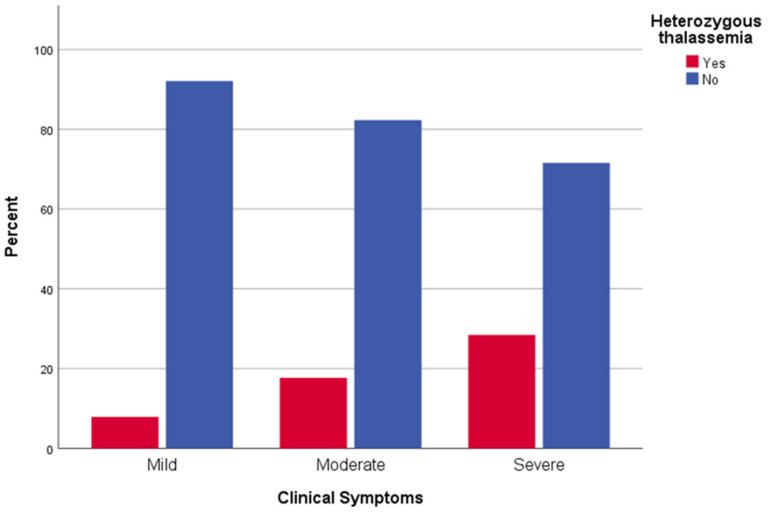
Clinical symptoms of COVID-19 in β-Thalassemia heterozygotes patients and control: distribution of clinical symptoms of SARS-CoV-2-infected β-Thalassemia trait carriers (red box) and non-carriers (blue box).

**Table 1 jpm-12-00352-t001:** Assessing mortality risk of study groups from COVID-19 by univariate and multivariate statistical analysis.

		Outcome: Mortality	Univariate			Multivariate Binary Logistic Regression	
		Yes (%)	Sig.	OR with 95% CI	RR with 95% CI	Sig.	aOR with 95% CI
Sex (M/F)	M: F:	140 (31.3) 49 (15.7)	**<0.001 (C)**	2.44 (1.69–3.51)	1.99 (1.49–2.66)	**<0.001 (C)**	2.59 (1.73–3.90)
Age (median, IQR)		Dead: 73 (16) Alive: 62 (23)	**<0.001 (M–W)**	-	-	**<0.001 (C)**	1.049 (1.031–1.066)
β-Thalassemia heterozygosity	Yes: No:	53 (38.7) 136 (21.8)	**<0.001 (C)**	2.26 (1.53–3.35)	1.77 (1.37–2.29)	**0.001**	2.41 (1.55–3.74)
Chronic respiratory disease	Yes: No:	41 (43.6) 148 (22.2)	**<0.001 (C)**	2.71 (1.73–4.23)	1.96 (1.50–2.57)	**0.018**	1.84 (1.11–3.05)
Atrial fibrillation	Yes: No:	84 (37.0) 103 (19.4)	**<0.001 (C)**	2.44 (1.73–3.45)	1.91 (1.50–2.43)	0.058	1.50 (0.99– 2.28)
Hypertension	Yes: No:	138 (32.9) 51 (15.0)	**<0.001 (C)**	2.77 (1.93–3.98)	2.19 (1.64–2.92)	0.243	1.31 (0.83–2.04)
Coronary disease	Yes: No:	53 (37.3) 136 (22.0)	**<0.001 (C)**	2.11 (1.43–3.12)	1.70 (1.31–2.20)	0.617	0.89 (0.55–1.42)
Diabetes mellitus type II	Yes: No:	48 (30.8) 141 (23.3)	0.056 (C)	1.46 (0.99–2.15)	1.32 (1.00–1.74)	0.439	0.84 (0.54–1.30)
Neoplasia	Yes: No:	27 (31.8) 162 (24.1)	0.122 (C)	1.47 (0.90–2.40)	1.32 (0.94–1.85)	0.653	0.88 (0.52–1.51)

C, Chi-square test; F, Fisher’s exact test; M–W, Mann–Whitney U test.

**Table 2 jpm-12-00352-t002:** Assessing clinical severity of COVID-19 by univariate and multivariate statistical analysis.

	Outcome: Severity			Univariate	Multivariate Ordinal Logistic Regression	
	Asymptomatic-Mild (%)	Moderate (%)	Severe-Critical (%)	Sig.	Sig.	aOR with 95% CI
Sex (Male)	94 (49.5)	210 (56.3)	144 (73.1)	**<0.001 (C)**	**<0.001**	1.98 (1.47–2.66)
Age (median, IQR)	52 (32)	65 (18)	72 (16)	**<0.001 (Κ–W)**	**<0.001**	1.052 (1.040–1.064)
Atrial Fibrillation	51 (26.8)	93 (24.9)	83 (42.6)	**<0.001 (C)**	0.373	0.85 (0.60–1.21)
Chronic respiratory disease	15 (7.9)	38 (10.2)	41 (20.8)	**<0.001 (C)**	0.098	1.45 (0.93–2.26)
Coronary disease	20 (10.5)	69 (18.5)	53 (26.9)	**<0.001 (C)**	0.634	1.10 (0.73–1.67)
Diabetes mellitus Type II	29 (15.3)	79 (21.2)	48 (24.4)	0.078 (C)	0.331	0.83 (0.58–1.20)
Neoplasia	20 (10.5)	35 (9.4)	30 (15.2)	0.108 (C)	0.173	0.73 (0.47–1.15)
Hypertension	69 (36.3)	204 (54.7)	147 (74.6)	**<0.001 (C)**	0.104	1.34 (0.94–1.91)
β-Thalassemia heterozygosity	15 (7.9)	66 (17.7)	56 (28.4)	**<0.001 (C)**	**<0.001**	2.59 (1.78–3.77)

C, Chi-square test; K–W, Kruskal–Wallis Test.

## Data Availability

Data are available upon reasoning request.

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
