# Peer review of "Vulnerability of β-Thalassemia Heterozygotes to COVID-19: Results from a Cohort Study"

_jpm, 2022, doi:10.3390/jpm12030352_

Round 1

Reviewer 1 Report

This study is interesting. However, there are several similar studies in the published literature. Despite this, the study may be publishable.

Comments:

Introduction
The introduction is very general. I suggest giving more details of the risk factors.

Methods
Study design not described: Is this a retrospective, cross-sectional or cohort observational study?
How were the 760 study participants selected?
What were the inclusion criteria?
Include the checklist STROBE.
Sample Estimate. A minimum study sample was set 89
at 231 patients?

Results:
Include the STROBE flowchart.
Table 1. Include comparator.
Lines 111-120 are repetitive of table 1. I suggest deleting these sentences.

Table 2. Multivariate Ordinal logistic regression. What variables are the three groups comparing?

What were the confounding variables in Tables 1 and 2? Figures 1 and 2 should perform analyzes adjusted for comorbidities, including ICU beds.

Discussion.
They should describe and discuss the limitations of the study.

Author Response

Reviewer’s concern: This study is interesting. However, there are several similar studies in the published literature. Despite this, the study may be publishable.

Authors’ answer: We would like to thank the reviewer for the positive comments. There are several studies associating mortality and morbidity due to COVID-19 with common comorbidities, however only one recently published pilot study from our center investigates the vulnerability of β-thalassemia heterozygotes to COVID-19.

Reviewer’s concern: Introduction

The introduction is very general. I suggest giving more details of the risk factors.

Authors’ answer: We would like to thank the reviewer for the well-stated comment. A new paragraph with details of risk factors associated with COVID-19 has been added in introduction section.

Reviewer’s concern: Methods

Study design not described: Is this a retrospective, cross-sectional or cohort observational study?

How were the 760 study participants selected?

What were the inclusion criteria?

Include the checklist STROBE.

Sample Estimate. A minimum study sample was set 89

at 231 patients?

Authors’ answer: We have adjusted the method section and the “patient” paragraph includes the following sentence “The present retrospective cohort study…” (Line 62). Our study population consists from consecutive patients during a six months period and all patients diagnosed positive by a PCR molecular conducted in the ER department test were included in the present study (Lines 63-66). Following your well-stated comment, the STROBE checklist was included in the supplementary material. Furthermore, we have conducted a power analysis using Epi-info, Considering an estimated prevalence of β-thalassemia heterozygosity at 8% in our entire study population (Thessaly, Greece), a precision of ±3.5% and 95% confidence interval (CI) and the minimum study population that were needed in order to identify a statistically significant difference was set at 231 patients.

Reviewer’s concern: Results:

Include the STROBE flowchart.

Table 1. Include comparator.                         

Lines 111-120 are repetitive of table 1. I suggest deleting these sentences.

Table 2. Multivariate Ordinal logistic regression. What variables are the three groups comparing?

What were the confounding variables in Tables 1 and 2? Figures 1 and 2 should perform analyzes adjusted for comorbidities, including ICU beds.

Authors’ answer:

Thank you for you comment. Our study design is a cohort study and does not include comparative data and using comparator cannot be applied. All consecutive patients during a six months period and all patients diagnosed positive by a PCR molecular conducted in the ER department test were included in the present study and there were no missing data or excluded patients. Thus, STROBE flow chart can not be applied in the present study. In table 1, in every variant, patients with the referring characteristic are comparing with the patients without the present characteristic. In Table 1 demographic characteristics as age and sex are presenting grouping by mortality and in the above-mentioned lines these characteristics are displayed in general regarding the whole population. Furthermore, the total mortality rate of the study is only displayed in the main text and not in a Table 1. Multivariate Ordinal logistic regression in Table 2, included all variants that were statistically significant in univariate analysis (age, sex, atrial fibrillation, chronic respiratory disease, coronary disease, hypertension and β-Thalassemia heterozygosity). Figure 1 represents the distribution of β-thalassemia heterozygotes among patients who died from COVID-19 and Figure 2 represents the distribution of β-thalassemia heterozygotes and clinical severity. As indicated in main text, the severe clinical symptoms are defined as the admission to ICU (Line 77-78). In Table 2 is displayed the adjusted analysis for comorbidities.

Discussion.

They should describe and discuss the limitations of the study.

Authors’ answer: The limitation of the present study are described in lines 204-210.

Reviewer 2 Report

The authors have assessed several risk factors in severity of COVID-19 including sex, age, and β-thalassemia. They investigated if these factors resulted in mortality as the first endpoint and a need for ICU as the second endpoint. Performing statistical analyses on 760 unvaccinated COVID-19 positive patients, in addition to sex and age factors, they have confirmed the previously published results identifying β-thalassemia as an important risk factor. The authors have discussed the basic science around their clinical findings. I believe the results would be of interest of basic science and clinical researchers as well as clinicians.

Author Response

We would like to thank the reviewer for the well-stated comments.

Round 2

Reviewer 1 Report

No comments